# Gym-Goers’ Self-Identification with Physically Attractive Fitness Trainers and Intention to Exercise

**DOI:** 10.3390/bs12050158

**Published:** 2022-05-23

**Authors:** Robert Jeyakumar Nathan, Vijay Victor, Tan Pei Kian

**Affiliations:** 1Faculty of Business, Multimedia University, Melaka 75450, Malaysia; soekma_wt@yahoo.com (S.); pktan@mmu.edu.my (T.P.K.); 2Department of Economics, CHRIST (Deemed to be University), Bhavani Nagar, Bangalore 560029, India; vijay.victor@christuniversity.in; 3College of Business and Economics, University of Johannesburg, Auckland Park 2006, South Africa

**Keywords:** sports psychology, self-identification process, wellbeing, fitness industry, sports marketing

## Abstract

Gym-goers often socially compare themselves with their trainers as they strive to look as attractive as their fitness trainers. The aim of this study was to better understand this phenomenon in the fitness industry. Relying on social comparison theory and social identity theory, self-identification with a physically attractive fitness trainer was posited to have a strong mediating effect on the relationship between appearance motive, weight management motive and gym-goers’ intention to exercise. The moderation effects of gym-goers’ age and gender in the direct relationships between appearance motive, weight management motive and exercise intention were also examined. The primary outcome of this study revealed that gym-goers who were influenced by appearance and weight management motives are more likely to identify with physically attractive fitness trainers. Additionally, gender significantly moderates the relationships between appearance motive, weight management motive and exercise intention. Appearance and weight management motives are the primary factors that influence the exercise intention of female gym-goers as compared to their male counterparts. This study sheds new insights into understanding the influence of the physical attractiveness of fitness trainers and its impact on gym-goers’ exercise intentions via self and social identification process.

## 1. Introduction

Physical inactivity and a sedentary lifestyle have been major health issues due to their incremental disadvantages to one’s wellbeing. This has been made worse by the recent COVID-19 crisis, which imposed home quarantines and restrictions on public activities and gatherings. Staying and working at home increases screen time and deskbound activity, resulting in decreased physical activity levels [1]. Moreover, it is common that physical activity declines at a high rate in adolescence and young adulthood [2,3]. Rowland [2] explained that the reasons for one’s declined physical activity can be intrinsic and biologically driven as “the result of a fall in central drive as well as other biological factors, such as decreasing skeletal muscle mass in older years” (p. 3). Besides biological and intrinsic barriers, extrinsic factors affecting physical activity levels are also influential. These extrinsic factors often navigate people away from physical activity in real-world situations. This is particularly true for females who are further stressed out by social pressure [2]. Additionally, convenient lifestyles and technological advancement indirectly encourage sedentary behaviours among young adults. Activities related to commuting, working and leisure involve significantly less physical activity now than before.

The positive and negative impacts of physical activity and inactivity are incremental in nature. “It has been recognised that most diseases affected by exercise are a result of life-long processes, surfacing clinically in older adult years” [2] (p. 2). A systematic review of longitudinal studies for the years 1996 to 2011 concluded a correlation between sedentary behaviour with a high risk of mortality in later ages, including those related to CVR, hypertension, symptomatic gallstone, site-specific cancers (ovarian, endometrial and colon), diabetes and even mental disorders [4]. A consistent relationship between sedentary behaviour during adolescence and weight gain or obesity in adulthood was also identified. Correspondingly, a sedentary lifestyle during adolescence would increase cardiometabolic risk in adulthood and vice versa [5].

Furthermore, a sedentary lifestyle can also lead to anxiety and depression [6,7]; when the level of physical activity falls, depressive symptoms experienced by a person increase [8]. Conversely, being physically active improves one’s mental health by reducing depressive symptoms. Being physically active during adolescence could also lower the risk of having depression symptoms throughout adulthood, up to the age of 62 [9]. This has given rise to recent research interest in health promotion and the wellbeing of employees at work, influenced by mental and physical health factors [10].

The isolation strategies to curb the COVID-19 spread restrict physical activity, leading to public anxiety and depression. Physical inactivity, decreased outdoor gathering and deteriorated mental health would negatively influence herd immunity, which is necessary for fighting the pandemic [11]. Correspondingly, sedentary behaviour and inactivity also lead to critical health issues in Malaysia. Physical inactivity results in several preventable illnesses diagnosed among Malaysians, including diabetes (18.3% of them), hypertension (30% of them) and hypercholesterolemia (38.1% of them) [12]. In addition, for the year 2019, Malaysia was listed as the top country with the highest overweight and obesity cases in Asia [13].

Understanding the need for reorienting today’s sedentary lifestyle, this study investigates how appearance and weight management motivations could influence gym-goers’ intention to exercise. There is a paucity of studies that quantitatively assess how the interactions between gym-goers’ appearance and weight management motives and self-identification with physically attractive trainers could lead to exercise intention. This empirical research attempts to fill this gap by examining the impact of appearance and weight management motives on gym-goers’ exercise intention, with the mediating effect of an identification process with physically attractive fitness trainers (Hypotheses 1 and 2). In addition, this study also investigates the moderating effect of gym-goers’ gender and age in the direct relationships between their appearance motive and exercise intention, as well as weight management motive and exercise intention (Hypotheses 3 and 4).

## 2. Literature Review and Theory

This study’s theoretical framework is based on three major psychological theories: self-determination theory, social comparison theory and social identity theory. Based on self-determination theory (SDT), motivation, also referred to as ‘the why of behaviour’, serves as why an individual performs an activity [14,15,16]. In day-to-day life, identifying the motivations of one’s action or behaviour is never straightforward; it is often caused by a combination of multiple inseparable motives operating as one entity. Motivations differ in their degree or level (how much motivation) and orientation (what type of motivation) a person exhibits. In the most basic SDT categorisation, motivation is divided into extrinsic motives and intrinsic motives. These two classifications of motives have been used in many psychological studies exploring ‘the why’ of various actions and behavioural intentions, including those about exercise intention. For instance, there are studies on physical activity and its motivations via self-determination theory [16], intrinsic versus extrinsic motivation in sport by Vallerand and Losier [14], extrinsic motivations in exercise adherence by Ednie and Stibor [17] and studies that incorporate a variety of motivational items that can be categorised into intrinsic or extrinsic motives [18,19,20].

The theory of the social comparison process, also known as social comparison theory, was established and developed by Leon Festinger [21]. In developing this theory, Festinger posited that a person chooses someone who has similar abilities or opinions for the purpose of comparison. Self-evaluation and social comparison naturally lead to a discrepancy in opinion and level of ability and cause pressure among members of a social group to reduce this difference. In the theory of the social comparison process, movements towards uniformity of abilities would be adequate by influencing members whose performance is lower and changing their value to be in line with their superior’s ability and, thus, increase their motivations to ‘drive upward’ [21,22]. Upwards comparison requires role models or agents to provide hope and inspiration who are often persuasive [22,23,24]. This new upwards direction led to the formulation of the new concepts of social comparison, including self-enhancement and self-improvement, which are significantly linked to assimilation to the upwards target. Assimilation has been previously studied under the idea that one could obtain the same upper status of the upwards target by psychological closeness, leading to the identification of the comparison target [23].

In social identity theory, ‘the self is reflexive. It can take itself as an object and categorise, classify, or name itself in particular ways concerning other social categories or classifications’ [25] (p. 224). In facilitating the social categorisation process, the social comparison process is prominent, especially for grouping those similar to the self as the in-group and those who are different from the self as the out-group. This process results in accentuating the perceived similarity among the in-group members and the perceived differences among the out-group members [25,26]. When a person defines themselves according to the social identity of a particular group, they attempt to align their behaviour with the group members’ behaviours [27]. In other words, self-categorisation or identification with a particular social group verifies one’s identity by looking similar to other members in the group and perceiving traits from the group’s perspective. Moreover, it also depersonalises an individual’s self-perception as one embodiment of the in-group prototype [25]. It assimilates “ones’ attitudes, feelings and behaviours to the in-group prototype” [26] (p. 187). Through depersonalisation, the in-group members develop the values of cohesiveness, inclusiveness, altruism and collectiveness, resulting in improved self-esteem and self-worth [25,28].

In sports psychology, social identity treats features of the sporting context as elements that can be integrated into one’s sense of self and turn them into a sport-related behaviour [29]. For instance, the members of a specific fitness group and fitness trainers mutually verify the features of ‘regular exercise’ and ‘healthy diet’ [30,31] and foster togetherness and belongingness among its members, and this often becomes the benchmark of their social identity and their prototype. When the newbies conform to the prototype and successfully verify their social identities, they are accepted as a part of this fitness social group, enhancing their self-esteem and leading to identification and assimilation of the group’s social identity as their one integral and communal identity [28]. Once the in-group identification is achieved, each member develops strong attraction, greater commitment and loyalty to the group [25].

Since identification fortifies an intragroup’s prototypicality, it would also concurringly support the influence possessed by the most prototypical member. As members cognitively and behaviourally follow the group’s prototype, they would automatically agree and comply with the verbal messages conveyed by the most prototypical member, creating a status-based differentiation between a leader and their followers [26]. Therefore, given that a ‘fit body’ is a prototype in the fitness group, a fitness trainer who is the most prototypical member (a member who has the fittest physique) is viewed as a leader who has the most influence on their group’s members.

### 2.1. Appearance and Weight Management Motives to Exercise

For the identification to happen, a gym-goers’ motivation to exercise must also be considered. Extrinsic motives to exercise refer to exercising motives that are driven by external reward and influence. They are generally described in terms of their fundamental characteristics subject to external and separable outcomes, whether a tangible reward, avoidance of punishment, avoidance of feelings of guilt/shame, attainment of recognition/approval or any combination of these characteristics [16]. Extrinsic motives to exercise are widely identified as the most common motives for novice exercisers or newbies [32]; they are also referred to as the most compelling motives for initiating exercise [33]. Among those extrinsic motives to exercise, health and appearance-related goals are repeatedly rated as the highest two [17,18,19,20].

Health has been used as an alibi and endorsement for appearance and weight management goals [30,34]. “A notion of health/fitness … was about maintaining a balance of food and exercise to achieve a slim body shape” [35] (p. 711). A physically fit and attractive fitness trainer is also perceived to be in their best health by gym-goers [36]. Due to the misleading association of health goal to weight management motive, in this study, health motive is excluded, leaving only two motives, i.e., appearance and weight management motives. In addition, the inclusion of appearance and weight management motives in this research is because these two motives are “central to personal trainers’ construction of the service that they offer to their clients” [30] (p. 27). Since these two exercise motives are the most highly voted motivations to exercise, in this study, it is posited that these two motives would have a direct impact on gym-goers’ exercise intention (Hypotheses 3 and 4).

Furthermore, prior research into exercise motivation suggests that gym-goers who are motivated by appearance and weight management goals are more subjected to self-objectification and have relatively low body satisfaction, body esteem and self-esteem, especially for women [31,37,38]. According to self-determination theory (SDT), appearance and weight management motives are categorised under external regulation and introjected regulation, the two most minor self-determined forms of extrinsic motivation. In an effort to improve their self-esteem and find a remedy for self-disapproval, appearance- and weight management-driven gym-goers were then predicted to socially compare and identify with a physically attractive fitness trainer and then persuade them to sign up for an exercise program offered (Hypotheses 1 and 2). An ample explanation of how identification with a physically attractive fitness trainer could persuade exercise intention of the appearance and weight management motives of gym-goers is given in the following subsection.

### 2.2. Gender Influencing Gym-Goers’ Preferences over Appearance and Weight Management Motives

Gender-specific characteristics could affect people’s behaviour, daily activities and relationships, including those within the physical activity domain. In the history of sports psychology, extrinsic exercise motives, especially those related to motives in appearance enhancement, attractiveness boost and health maintenance, have been associated with women [37,39]. A study of college students’ motivation for physical activity identified that college women had “greater concerns regarding their body weight than men” [18] (p. 93). Women with higher body mass index (BMI) would exercise to lose weight [40]. Polman [41] also supported the finding that women’s body composition affected their satisfaction, making them prefer weight management as their top exercise motivation. Women are 2.4 times more likely than men to choose appearance goals as their exercise motive [33]. As for men, they are more likely to view physical activity as a means to pursue ego-related outcomes, such as their show of mastery and competence. “Men reported higher levels of motivation … for a challenge, competition, social recognition, and strength and endurance” [18] (p. 54). Similarly, Ednie [17] found that men’s higher ratings on their exercise motives are enjoyment, challenge, competition and social motives.

Women’s appearance and weight management motives are related to what society and culture perceive as a beauty standard—women’s beauty is equal to a slim and slender figure. Society and culture pressure young women to be overly concerned about their body size and weight. Moreover, “females are highly influenced by others, including society, and their gym participation is due to pressure as opposed to desire” [37] (p. 13). Similarly, Anić et al. [40] found that women influenced by appearance and weight management motives are significantly influenced by sociocultural pressure as well as appearance idealisation [40]. Women tend to be more motivated by appearance and weight management motives, often because of their high motivation, low perceived competence, low task orientation and low incremental beliefs [42]. As a result, they resort to external stimulation such as fitness trainers’ physical attractiveness as their motivation to exercise [32,43]. Hypothetically, given that women would be more motivated by appearance and weight management motives, they would be more possibly convinced to sign up for an exercise program offered. Thus, this empirical study predicted that gym-goers’ gender would moderate the direct relationships between appearance motive and gym-goer’s exercise intention (Hypothesis 3a) and weight management motive and gym-goer’s exercise intention (Hypothesis 4a).

### 2.3. Age Influencing Gym-Goers’ Preferences over Appearance and Weight Management Motives

When it comes to exercise motivation, various prior studies have focused on exercise adherence besides exercise intention [14,16,17,18,44]. It was found from preliminary research that young adults appreciate and value physical attractiveness substantially more than middle-aged adults and elderly adults. Thus, they are more motivated by appearance and weight management goals as ways to enhance their own physical attractiveness.

Furthermore, Caglar et al. [20] found that motives related to appearance are one of the most important motives for exercise among participants aged 21 to 24-years-old. In line with this, Kilpatrick et al. [18] also discovered that among the top three motives of college students for physical activity, appearance and weight management motives dominated them. In the study of Brunet and Sabiston [44], a significant positive correlation was also found between introjected regulation and physical activity for young adults, reflecting on the body image concerns and self-imposed pressure in obtaining or maintaining a desired physical appearance among young adults.

The elderly care more about their health, mental wellbeing and extending their lives [33,45] than their appearance and physical attractiveness. This motivational pattern related to the elderly is also supported by a qualitative study by Rotwein [46]. The study found that people above 40 years old would strive to improve their quality of life and stay healthy. Appearance objectives such as losing weight and becoming toned are no longer their main focuses. Theoretically, young adults are more motivated by appearance and weight management motives and, hence, could be easily persuaded to sign up for an exercise program offered. Therefore, this study posited that gym-goers’ age would moderate the direct relationships between appearance motive and gym-goer’s exercise intention (Hypothesis 3b) and weight management motive and gym-goer’s exercise intention (Hypothesis 4b).

### 2.4. Self-Identification with Physically Attractive Fitness Trainers

A great physique of a trainer is a reflection of the professionalism and their projection as a good role model and exemplar [31]. In a fitness group, a great physique of a physically attractive trainer represents their quality as the most prototypical member who has an influential impact on other members. This is because members of a particular fitness centre would attempt to conform to this prototype of a fit physique. This justification is also supported by prior studies that found that gym-goers’ preferred body type is often similar to their instructor’s figure, indicating they wish to have a physique similar to their instructor’s body [32,43,46,47].

Gym-goers’ identification with a physically attractive fitness trainer begins with socially comparing themselves with a physically attractive trainer. The social comparison process in the fitness environment is related to evaluating ability that would instigate self-improvement motive; this comparison process is also known as an upwards comparison. In the study on social comparison among fitness app users, upwards fitness comparison with higher performers elevates users’ self-efficacy and motivation to exercise, ultimately resulting in their participation in physical activity [48]. An alterable state of one’s appearance would cause a temporary improvement in self-perception, self-esteem [49] and self-efficacy [48]. In other words, if gym-goers believe that they could improve their appearance to compete with their physically attractive fitness trainer, the comparison can be uplifting, and their reaction to the comparison would not be negative. “The more easily improved a body part is perceived to be, the less likely a comparer is to feel negatively as a result of the comparison” [49] (p. 61). After comparing themselves with a physically attractive trainer, gym-goers would identify with the trainer as their idol and role model. Identification is motivated by the disinhibitory effect, whereby gym-goers repeat a similar action or exercise regime given by a fitness trainer expecting that they can have an attractive physique and fitness appearance of the trainer.

Interestingly, social comparison to a superior other, including a physically attractive fitness trainer, evokes self-inferiority and self-dissatisfaction. Exposing gym-goers to a physically attractive fitness trainer as the attractive model would lower their self-image and evoke a feeling of dissatisfaction while simultaneously presenting the product (exercise program offered) as a relief or remedy from those negative emotions [50]. Self-inferiority and dissatisfaction, triggered by social comparison to a physically attractive fitness trainer, are more likely to be gym-goers driven by appearance and weight management motives. Gym-goers’ incentives for seeking a trainer often originated from the negative force due to their frustration because they failed to attain their desired physique or fitness appearance [47]. Through this identification process, gym-goers who are driven by appearance and weight management motives could be convinced to sign up for an exercise program, leading to their exercise intention (Hypotheses 1 and 2).

Based on the literature review and theoretical discussions above, a theoretical framework was constructed (Figure 1). This framework contains Hypotheses 1 to 4 (H1 to H4), listed as follows.

**Hypothesis** **1** **(H1).***Gym-goers’ identification with physically attractive fitness trainer mediates the relationship between appearance motive and exercise intention*.

**Hypothesis** **1a** **(H1a).***Appearance motive of gym-goers influences their identification with physically attractive trainer*.

**Hypothesis** **1b** **(H1b).***Gym-goers’ identification with physically attractive fitness trainer influences their exercise intention*.

**Hypothesis** **2** **(H2).***Gym-goers’ identification with physically attractive fitness trainer mediates the relationship between their weight management motive and exercise intention*.

**Hypothesis** **2a** **(H2a).***Weight management motive of gym-goers influences their identification with physically attractive trainer*.

**Hypothesis** **2b** **(H2b).***Gym-goers’ identification with physically attractive fitness trainer influences their exercise intention*.

**Hypothesis** **3** **(H3).***Appearance motive of gym-goers influences their exercise intention*.

**Hypothesis** **3a** **(H3a).***Gym-goers’ gender moderates the relationship between their appearance motive and exercise intention*.

**Hypothesis** **3b** **(H3b).***Gym-goers’ age moderates the relationship between their appearance motive and exercise intention*.

**Hypothesis** **4** **(H4).***Weight management motive of gym-goers influences their exercise intention*.

**Hypothesis** **4a** **(H4a).***Gym-goers’ gender moderates the relationship between their weight management motive and exercise intention*.

**Hypothesis** **4b** **(H4b).***Gym-goers’ age moderates the relationship between their weight management motive and exercise intention*.

## 3. Materials and Methods

This study investigates how gym-goers’ appearance and weight management motives influence their exercise intention, mediated by identification with a physically attractive fitness trainer, moderated by age and gender. The theoretical framework in Figure 1 depicts the postulated relationship among the research variables. In this quantitative cross-sectional study, a survey questionnaire was used as the research instrument to collect the research data. The questionnaire used photographs of a physically attractive fitness trainer, labelled ‘Dave’, as its visual stimuli. In addition to the visual stimuli, the questionnaire also includes textual scenarios introducing the setting and facilities of the hypothetical gym and a description of the exercise program, which served as verbal apparatuses that allowed the respondents to envision a real-world setting.

‘Dave’ or Model 3 was the highest-rated model among the other three models tested in the Preliminary Attractiveness Survey (PAS); 56 per cent of the respondents rated ‘Dave’ as the most attractive model. A single written instruction: ‘Please rate the highest attractive model in terms of his physique and facial attractiveness’ was asked to all respondents of the PAS. Three photos displaying attractiveness of both face and physique (head to knee) of anonymous, shirtless models, labelled as Model 1, 2 and 3, were presented for respondents’ selection in the PAS. Although physical attractiveness can be ‘wrought’ using make-up, this is almost impossible in the fitness world since it involves physically intense and sweaty activities making cosmetic make-up ineffective. The physical attractiveness of a fitness trainer is more intricate, as it is not only measured by mainly pleasant facial appearance alone but also the combination of a lean and defined body figure. In the fitness industry, a trainer’s fit physique is part of their physical attractiveness. “A source can be attractive and not physically fit, yet, if a source is perceived as physically fit, this would likely increase the overall perceived attractiveness” [51] (p. 15). The lucrative value of physique and physical appearance attached to trainers is also referred to in recent studies as “bodily capital”. In a critical narrative analysis of the perspectives of fitness trainers in relation to their body image, it concluded that a trainer’s physique was treated as a commodity and also served as physical capital [31].

All items under the six variables are measured on a five-point Likert scale ranging from (1) ‘strongly disagree’ to (5) ‘strongly agree’. Except for the mediating variables, the scale items were adapted from previous studies on physical attractiveness, exercise motivation, perceptions of fit physique and social comparison/identification with a prototypical leader. These items were cross-checked with similar items (if any) utilised by recent studies on bodily capital in the fitness industry and the identification process within exercising and sports activities. For previous studies that are qualitative and inductive in their approach, frequently mentioned keywords and terms were identified and synthesised in order to construct additional scale items in the questionnaire. At the end of this process, a total of 21 items or indicators were finalised. Table 1 shows the measurement items and their relevant sources.

A total of 214 samples were collected from gym-goers of 10 randomly selected fitness gyms across three districts of the Melaka State (Central Melaka, Jasin and Alor Gajah) in Malaysia. The full list of all privately owned gyms in Melaka was obtained from the Companies Commission of Malaysia (Suruhjaya Syarikat Malaysia). This list was then combined with the list provided by the Melaka Department of Youth and Sports (Jabatan Belia dan Sukan Melaka), which consisted of government-owned gyms in Melaka. The combined list was again cross-checked with the list available online to derive a final list consisting of all gyms in Melaka. Ten gyms were randomly selected from the list via the INDEX function in Microsoft Excel with the help of ‘RANDBETWEEN’ and ‘ROWS’ functions. The number of gyms chosen for each of these three districts of Melaka State was calculated using the proportional ratio, based on the quantity of actual gyms located in each district. A breakdown of the calculation procedure is shown in Table 2.

For every gym, 22 members were randomly selected from its members’ list using the Index function in Excel. The chosen respondents were invited to participate in the survey. Out of the 10 gyms, 5 gyms responded and agreed to participate in the study. Among the five participating gyms, the participation rate was 95% for four gyms, while one gym had a 91% participation rate. Overall, participants from the gyms selected were cooperative and voluntarily participated in the survey. Upon gathering raw data from 214 samples, the individual samples were assigned unique IDs. Further, the data were organised in the Excel file and then extracted into the IBM SPSS Software, Version 27, for further analysis. There were no missing observations in the final dataset. Extreme outliers in the data were detected using boxplots. The deletion of extreme outlier samples was performed step by step by research variable. The final dataset consisted of 192 samples finalised for descriptive and hypothesis testing.

## 4. Results

Table 3 shows the respondents’ demographic information, organised and segregated based on age and gender. The respondents were primarily young adults between 18 to 40 years old. The number of male and female participants was 95 and 97, respectively. An independent sample *t*-test to compare means for the responses towards the physical attractiveness of the fitness trainer between male and female groups was carried out to identify potential differences between the two gender groups. Upon running the *t*-test, no significant differences (*p*-value = 0.547) between males and females in terms of their responses to the physical attractiveness of the fitness trainer were found.

As displayed in Table 4, all items had outer loadings above 0.70, except AM2, AM5, WM2 and WM4. Factor loadings below 0.40 are recommended to be removed from the model [59] due to their poor relationship with their construct. Factor loadings of 0.50 and higher that are statistically significant could be considered relevant in measuring their construct [60]. Out of 21 indicators, none had a factor loading below 0.40. The four indicators (AM2, AM5, WM2 and WM4) with loadings below 0.70 were higher than 0.40. These indicators were retained in the final model, as their constructs’ composite reliability was above 0.70, and their constructs’ AVEs were all above 0.50. Hence, all the indicators and constructs were finalised for hypothesis testing, as they met the convergent validity criteria [61,62].

Cronbach’s alpha of each construct was assessed to check for internal consistency. The acceptable standard cut-off for Cronbach’s alpha is 0.70 [63,64]. The results in Table 5 show that Cronbach’s alpha is higher than the threshold level of 0.70. Convergent validity is confirmed by assessing the factor loadings, composite reliability and average variance extracted (AVE). Based on the results, all four variables, AM, WM, IPA and EI, satisfy the criteria.

Table 6 shows that the square root of the AVE for each variable is larger than the correlation of the respective variable to the remaining variables. In addition, the HTMT values shown in Table 7 are all less than the threshold of 0.85 [65,66]; the upper and lower boundaries of their confidence intervals do not include the result of 1, thus indicating no issues related to discriminant validity.

The hypotheses test results are presented in Table 8. Out of all hypotheses and their sub hypotheses, seven are supported. Results of all path coefficients and corresponding *p*-values are presented below in Table 8.

Both appearance and weight management motives significantly affect gym-goers’ self-identification with physically attractive fitness trainers (Hypotheses 1a and 2a are supported). Moreover, self-identification with physically attractive fitness trainers also significantly influences gym-goers’ exercise intention (Hypothesis 1b is supported). These findings show only indirect mediation effects via gym-goers’ identification with physically attractive fitness trainers in the relationships between the two motives and exercise intention (Hypotheses 1 and 2 are supported). These mediation effects are reconfirmed with the significant *p*-values for both their total indirect effects (AM → EI and WM → EI) as well as specific indirect effects (AM → IPA → EI and WM → IPA → EI). The path coefficients and *p*-values in Table 9 and Table 10 are the same, as there is only one mediating variable (IPA) in the model. In other words, the parallel values indicate that the total indirect effects are purely based on a single mediating variable that affects the two specific indirect relationships. There are no significant, direct relationships between AM and EI and WM and EI (Hypotheses 3 and 4 are not supported).

In addition, the moderating effects of gym-goers’ gender and age in the direct paths of AM → EI and WM → EI were also tested. It was found that only gender had a significant moderating effect on the relationship between appearance and weight management motives and exercise intention (Hypotheses 3a and 4a are supported). Moreover, the moderating effect of age was insignificant, indicating that Hypotheses 3b and 4b are not supported. In order to better understand how these two groups of ‘male and female gym-goers’ and ‘18–40 and 41–64 years old gym-goers’ differed, a multigroup analysis (MGA) was performed. A complete bootstrapping with 5000 iterations was conducted for both groups. The results of the MGA for gender, as presented in Table 11, show that the path coefficients and *p*-values for the relationships of AM to EI and WM to EI differed significantly. These results suggest a strong moderating effect of gender in the direct relationships between the two motives and exercise intention, as predicted in Hypotheses 3a and 4a.

Referring to the MGA results for the two age groups, as shown in Table 12, the differences in path coefficients for interactions from WM → EI, and WM → IPA are profound. The effect of age as a moderator (weight management motive*age) in the relationship between weight management motive and exercise intention is insignificant. Since the differences in path coefficients for interactions from AM → EI, and AM → IPA are insignificant between the two age groups, it is concluded that age as a moderator had a significant impact only at the 90% confidence level in the relationship between WM and IPA. However, this study’s structural model intended to investigate the moderating effects in the direct relationship between AM to GEI, and WM to EI, precluding any interaction towards IPA.

The path coefficients for the direct relationships between AM → EI and WM → EI are insignificant (please refer to H3 and H4 in Table 7). The moderating effects of gender (gender*appearance motive and gender*weight management motive) on the direct relationships (AM → EI and WM → EI) are significant. These peculiar findings are possible due to a crossover interaction, also known as disordinal interaction [67]. Crossover interaction happens when the path coefficients of interactions in the two subsamples are in opposite directions [68]. Since one path coefficient is positive while another is negative, they would average the main effect, causing it to be insignificant. Similarly, the path coefficients for the relationships of AM → EI and WM → EI for male and female groups (shown in Table 13 in bold) have opposite signs. Therefore, they cancel out each other, causing the main effects of AM → EI and WM → EI to be insignificant.

Further analysis of the MGA correspondingly reveals the significant differences between male and female groups in path coefficients for AM → EI, and WM → EI relationships. Moreover, the path coefficients for males are both in negative values, indicating negative relationships between AM → EI and WM → EI. In contrast, the path coefficients for females are both positive, indicating positive associations between AM → EI and WM → EI for female gym-goers.

In addition to the convergent and discriminant validity check, assessments of the model’s predictive relevance (Table 14) and goodness of fit were also carried out. Values of the predictive relevance (Q2) for all endogenous constructs were obtained via the blindfolding procedure run in the SmartPLS Software. As long as the values of Q2 are greater than zero or a positive value, the model is deemed to have predictive performance. In this model, all the Q2 were greater than zero, indicating predictive relevance for both gym-goer’s exercise intention and gym-goer’s identification with physically attractive trainer.

To enhance the inference derived from the result of predictive relevance, the Goodness of Fit (GoF) Index was calculated, as shown in the following table. The GoF Index is the geometric mean of the average communality (AVE) and average coefficients of determination (R2). “It can be used to determine the overall prediction power of the large complex model by accounting for the performance of both measurement and structural parameters” [69] (p. 4). Moreover, the GoF Index is suitable for both reflective and formative latent variables, even for a multifaceted model [70]. The model is said to fulfil the GoF criteria if the GoF Index obtained surpasses the cut-offs of 0.10 for a sample with a small effect size: 0.25 for a sample with a medium effect size and 0.36 for a sample with a large effect size [62,69]. Similarly, shown in Table 15, Goodness of Fit (GoF) Index, the GoF Index for this structural model was 0.3315 and, therefore, the criteria for the GoF Index in this study were met, as it was above the baseline of 0.25.

## 5. Discussion

Based on the demographic analysis, the descriptive analysis showed that exercising in gyms is more preferred by young adults in the study. The elderly may opt for other exercise alternatives, such as leisure-time physical activity, including outdoor walking, gardening and dancing. This finding corresponds with the result of Poveda-López et al. [71], who found the elderly mostly prefer outdoor exercise, especially walking sessions.

The main finding of this study showed that gym-goers motivated by appearance and weight management motives would identify with a physically attractive fitness trainer and eventually be persuaded to sign up for an exercise program, leading to exercise intention. The results of previous studies support this finding. Since appearance- and weight management-motivated gym-goers were previously found to have self-objectification, low body satisfaction and low self-esteem [31,37,38,47], a physically attractive trainer could motivate them to join an exercise program offered. In another study conducted by Apaolaza-Ibáñez et al. [50], the exercise program offered would be perceived as a remedy for individuals with negative perceptions about themselves. This is because gym-goers’ social comparison to a physically attractive fitness trainer is an upwards comparison that instigates self-improvement motives related to their appearance. The possibility for gym-goers to alter their physiques to their desired physical appearance would ultimately lead to their identification with a physical attractive fitness trainer as their role model [32,43], motivation [43,46,47,51] and fitness leader [72].

Moreover, appearance- and weight management-motivated gym-goers are more likely to socially compare and identify with a physically attractive fitness trainer because they wish to have a physique similar to their trainer’s body [32,43,46,47] Socially comparing one physique with another would be instantaneous, as physical attractiveness is a peripheral feature that is assessed externally. Gym-goers motivated by appearance and weight management motives are more likely to compare their physique with that of a physically attractive fitness trainer because an aesthetic and/or a fit physique is their main focus and goal for exercising in a gym. Having an attractive physique as their aspiration would ultimately lead them to assimilation and identification with a physically attractive fitness trainer because gym-goers see a physically attractive appearance of a fitness trainer as an in-group prototype that they would like to conform to within a ‘gym’ social group.

Another unique finding of this study is the moderating effect of gym-goers’ gender and age. Results of the data analysis revealed that only gender significantly moderates the relationships between the two exercise motives and exercise intention. In addition, the path coefficients of AM to EI and WM to EI for males and females were negative and positive, respectively. This could be due to low ratings on appearance and weight management motives by male respondents as opposed to female respondents, who rated appearance and weight management motives highly as important to them. Comparing this finding with the prior literature, such a pattern was predicted initially, which found a predisposition for female exercisers to be driven by appearance and weight management motives [18,37,39,41]. As for males, they were driven by motivations related to challenges, competition, social recognition, strength and endurance [17,18].

Hypothetically, women would be more inclined to identify with a physically attractive fitness trainer due to their preferences of appearance and weight management motives. However, the gender-wise moderation effect was limited to the direct effects of exercise motives on gym-goers’ exercise intention. As for confirming the crossover interaction detected, future studies could test this study’s theoretical framework on female or male gym-goers in two separate studies.

Based on the findings of this study, the age of gym-goers does not moderate the impact of AM to EI and WM to EI. This result indicates that older gym-goers are perhaps still concerned about their appearance and weight management when going to the gym to exercise. This could be because the elderly who exercise in a gym consist of individuals motivated by appearance and weight management motives; the elderly who are motivated by health-related motives may choose to engage in physical activities outside the gym facility.

Besides its theoretical implication, the findings of this study suggest an efficient strategy for gyms in marketing fitness as a lifestyle. Gyms hiring physically attractive trainers and employing them as role models in selling fitness programs in gyms is a step in the right direction for gyms. Additionally, fitness trainers and gyms should be able to customise their approach in selling their exercise programs according to the gender of their potential clients, as their motivations may differ.

Beyond its direct managerial implication to fitness trainers and gyms, this study would benefit the public sector, educational institutions, health philanthropists and also international organisations interested in the societal wellbeing agenda, as this study, in general, supports the promotion of an active lifestyle amidst the growing sedentary lifestyle prevalent especially among young adults.

## 6. Limitations and Future Study

This study suffers several limitations despite achieving its main objective. Firstly, the dependent variable exercise intention is measured as a behavioural intention and not actual action. Although behavioural intention is the main index of one’s psychological readiness for their action, there is still a non-negligible gap between intention and behaviour [73]. Thus, the findings of this study may not be directly generalised to actual exercise action.

Utilising a scenario-based questionnaire with photographs would intensify the effect of physical attractiveness, as it has lesser distractions than actual occurrences. This limitation was also found in Bashi et al. [74]. The respondents paid better attention to the physical attractiveness of the service contact personnel in photos than in real life. Furthermore, “the evaluative implications of looks may be somewhat weak in many natural settings, in which perceivers have extensive information about other people and their physical appearance. Individuating information would attenuate inferences made based on physical attractiveness; the stereotype was weaker when such information was present rather than absent” [75] (p. 122, p. 119). Hence, physical attractiveness could be less effective in a real-life setting, as gym-goers would be able to evaluate their trainers beyond their peripheral cues. Thus, the results of this study and findings interpretation are applicable on a first-impression basis, which relies significantly on peripheral (outer) cues.

Future studies could investigate this matter further by scrutinising the reasons for and perception of exercising in the gym. This could help to firstly determine gym-goers’ motivation to exercise. It is also worthwhile to explore older adult gym-goers’ perception of health-related motives to determine whether such motives are associated with appearance and weight management motives.

Additionally, future studies could study the next steps that occur after the first impression of a gym-goer towards a trainer, which includes investigating whether the physical attractiveness of fitness trainers could encourage sustainability of a fitness lifestyle and prevent drop-offs. Further studies could include motivational factors to explore strategies for young adults to lead an active lifestyle, such as their conception of sporting ability beliefs, perceived competence and behavioural regulations.

## 7. Conclusions

The findings of this study contribute to the existing body of knowledge, especially in the area of exercise psychology. This research is among the few to quantitatively study the indirect effects of self-identification with physically attractive fitness trainers and gym-goers’ exercise intentions. The advantages and effectual power possessed by a physically attractive person have been observed and recognised in various social and business functions. In this study, both appearance and weight management motives of gym-goers lead to their exercise intention via the mediating effect of self-identification with a physically attractive fitness trainer. In addition, the moderating effect of gym-goers’ gender is also identified in the direct relationships between these two motives and gym-goers’ exercise intention, in which they interact in a crossover manner. Thus, this study concludes that gym-goers driven by appearance and weight management motives could potentially be persuaded to start an active lifestyle via the effect of the self-identification process with a physically attractive fitness trainer.

## Figures and Tables

**Figure 1 behavsci-12-00158-f001:**
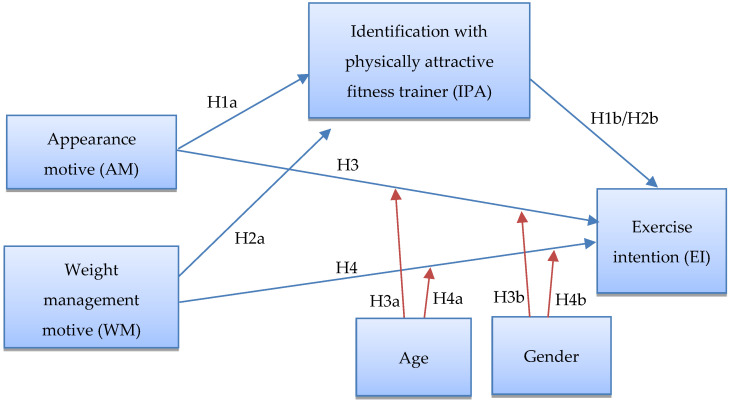
Theoretical framework.

**Table 1 behavsci-12-00158-t001:** Scale items and their sources.

Variable (Indicators)	Scale Items (Keywords)	Sources
AM (AM1, AM2, AM3, AM, AM5)	To improve appearance and obtain good, attractive, younger and toned look	Adapted from [17,18,19,20,39,40,52]
WM (WM1, WM2, WM3, WM4, WM5)	To look slim, lose weight, control weight, burn excessive calories and maintain weight	Adapted from [18,19,33,39,40,52]
IPA (IPA1, IPA2, IPA3, IPA4, IPA5, IPA6)	Role model, great exemplar, trainer represents important values and goals, strive to look similar to trainer, have an ideal physique comparable to trainer’s body and feel similar to trainer	Adapted from [31,32,43,46,47,51,53,54]
EI (EI 1, EI2, EI3, EI4, EI5)	Inquire, obtain trainer’s contact number (self-developed), receive a trial session (self-developed), consider signing up, would sign up	Adapted from [55,56,57,58]

Abbreviations: AM, appearance motive; WM, weight management motive; IPA, gym-goers’ identification with physically attractive fitness trainer; EI, gym-goers’ exercise intention.

**Table 2 behavsci-12-00158-t002:** Calculation procedure for selection of gyms.

District	Actual Number	Number Selected
Melaka Tengah	30 gyms	30/39 × 10 = 7 gyms
Alor Gajah	6 gyms	6/39 × 10 = 2 gyms
Jasin	3 gyms	3/39 × 10 = 1 gym
Total	39 gyms	10 gyms

**Table 3 behavsci-12-00158-t003:** Respondents’ demographic information.

Demographics	Category	Frequency	Percentage
Age Group	18–40 years old	152	79.2%
	41–64 years old	40	20.8%
Gender	Female	97	50.5%
	Male	95	49.5%

**Table 4 behavsci-12-00158-t004:** Factor loadings.

Variable/Construct	Indicator	Factor Loading
Gym-goers’ exercise intention	EI1	0.800
	EI2	0.835
	EI3	0.846
	EI4	0.912
	EI5	0.847
Gym-goers’ identification with physically attractive fitness trainer	IPA1	0.796
	IPA2	0.713
	IPA3	0.809
	IPA4	0.882
	IPA5	0.814
	IPA6	0.845
Appearance motive	AM1	0.789
	AM2	0.694
	AM3	0.748
	AM4	0.862
	AM5	0.525
Weight management motive	WM1	0.783
	WM2	0.658
	WM3	0.779
	WM4	0.697
	WM5	0.708

**Table 5 behavsci-12-00158-t005:** Construct reliability and validity.

Variable	Cronbach’s Alpha	Composite Reliability	AVE
AM	0.840	0.886	0.610
WM	0.923	0.942	0.764
IPA	0.858	0.895	0.589
EI	0.903	0.929	0.723

**Table 6 behavsci-12-00158-t006:** Fornell–Larcker criterion.

ID	Construct	1	3	5	6
1	AM	0.732			
2	EI	0.299	0.849		
3	IPA	0.441	0.687	0.811	
4	WM	0.470	0.281	0.373	0.730

**Table 7 behavsci-12-00158-t007:** Heterotrait–Monorait ratio (HTMT).

ID	Construct	1	2	3	4
1	AM				
2	EI	0.355 (0.222; 0.483)			
3	IPA	0.520 (0.401; 0.625)	0.758 (0.668; 0.829)	0.163 (0.105; 0.213)	
4	WM	0.600 (0.473; 0.714)	0.322 (0.208; 0.436)	0.245 (0.135; 0.341)	0.443 (0.316; 0.563)

Note: The values in parentheses denote the lower and upper boundaries for the 95% confidence interval.

**Table 8 behavsci-12-00158-t008:** Result of hypotheses testing.

Hypothesis	Relationship (Hypothesis Statement)	Path Coeff	*p*-Value	Decision
H1	Gym-goers’ identification with physically attractive fitness trainer mediates the relationship between appearance motive and exercise intention			Supported
H1a	Appearance motive → Gym-goers’ identification with physically attractive fitness trainer	0.342	<0.001 **	Supported
H1b	Gym-goers’ identification with physically attractive fitness trainer → Exercise intention	0.450	<0.001 **	Supported
H2	Gym-goers’ identification with physically attractive fitness trainer mediates the relationship between weight management motive and exercise intention			Supported
H2a	Weight management motive → Identification with physically attractive fitness trainer	0.212	0.002 **	Supported
H3	Appearance motive → Exercise intention	0.020	0.402	Not supported
H3a	Moderating effect 2 (appearance motive, gender) → Exercise intention	0.139	0.027 *	Supported
H3b	Moderating effect 1 (appearance motive, age) → Exercise intention	−0.009	0.449	Not supported
H4	Weight management motive → Exercise intention	−0.066	0.151	Not supported
H4a	Moderating effect 4 (weight management motive, gender) → Exercise intention	−0.120	0.036 *	Supported
H4b	Moderating effect 3 (weight management motive, age) → Exercise intention	−0.064	0.199	Not supported

Note: ** Significant at 1%; * significant at 5%.

**Table 9 behavsci-12-00158-t009:** Total indirect effects.

Relationship	Path Coefficient	T Statistics	*p*-Values
AM → EI	0.154	3.478	<0.001 *
WM → EI	0.095	2.385	0.009 *

Note: * Significant at 1%.

**Table 10 behavsci-12-00158-t010:** Specific indirect effects.

Relationship	Path Coefficient	T Statistics	*p*-Values
AM → IPA → EI	0.154	3.478	<0.001 *
WM → IPA → EI	0.095	2.385	0.009 *

Note: * Significant at 1%.

**Table 11 behavsci-12-00158-t011:** Multigroup analysis for gender.

Relationship	Path Coefficients—Diff (|Females–Males|)	*t*-Value (Females v. Males)	*p*-Value (Females v. Males)
AM → EI	0.350	2.150	0.016 *
AM → IPA	0.023	0.172	0.432
WM → EI	0.250	1.958	0.026 *
WM → IPA	0.093	0.639	0.262

Note: * significant at 5%.

**Table 12 behavsci-12-00158-t012:** Multigroup analysis for age groups.

Relationship	Path Coefficients—Diff (18–40 v. 41–64)	*t*-Value (18–40 v. 41–64)	*p*-Value (18–40 v. 41–64)
AM → EI	0.123	0.592	0.277
AM →IPA	0.118	0.725	0.235
WM → EI	0.310	1.817	0.035 **
WM → IPA	0.267	1.561	0.060 *

Note: ** significant at 5%; * significant at 10%.

**Table 13 behavsci-12-00158-t013:** Path coefficients comparison between gender groups.

Relationship	Path Coefficient Original (Females)	Path Coefficient Original (Males)	*p*-Value (Females)	*p*-Value (Males)
AM → EI	0.244	−0.106	0.016 *	0.178
AM → IPA	0.310	0.333	0.002 **	<0.001 **
WM → EI	0.052	−0.198	0.017 *	0.275
WM → IPA	0.291	0.198	0.004 **	0.019 *

Note: ** Significant at 1%; * significant at 5%.

**Table 14 behavsci-12-00158-t014:** Predictive relevance.

Endogenous Construct	Q²
Gym-goers’ exercise intention	0.363
Gym-goers’ identification with physically attractive trainer	0.139

**Table 15 behavsci-12-00158-t015:** Goodness of Fit (GoF) Index.

Construct	AVE	R2
Gym-goer’s identification with physically attractive trainer	0.658	0.230
Gym-goer’s exercise intention	0.720	0.555
Averaged AVE/R2	0.8445	0.3925
GoF = 0.8445 × 0.3925	0.3315	

## Data Availability

The data presented in this study are available on request from the corresponding author.

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
