# Peer review of "Gym-Goers’ Self-Identification with Physically Attractive Fitness Trainers and Intention to Exercise"

_behavsci, 2022, doi:10.3390/bs12050158_

Round 1

Reviewer 1 Report

First of all, thank you for allowing me to make some comments on this article whose subject I consider helping the scientific community to improve on this subject, which I consider to be of great interest.

First, I think it would be necessary to update both the introduction and the theoretical framework of the references used. Physical activity and the sports area are scientific disciplines that have a lot of evidence and great academic production in recent years that will undoubtedly provide more up-to-date references to this article. This issue is repeated throughout the article, so a complete review and update of references is requested.

With respect to the method, it would be interesting to know how the sample was selected, perhaps there are tables that are of little interest given the comments that derive from them, it being possible to consider removing them, the same thing happens with the results section, at times the information described is repeated with which appears in the tables.

In the discussion, the authors are again requested to update the measurement of possible references and adapt the entire article with the tables and references to the format of the journal.

I consider it of great interest that the authors add a section on the implications of the study and limitations, since I consider that it is necessary for the reader to know the application of the work as well as how to face the limitations it presents.

Author Response

Thank you esteemed reviewer. 

Cover page with Errata Sheet: Responses to Reviewer 1’s Comments and Corrections

Article Title: “Gym Goer’s Identification to Physically Attractive Fitness Trainer and their Intention to Exercise”

Reviewer #1

First of all, thank you for allowing me to make some comments on this article whose subject I consider helping the scientific community to improve on this subject, which I consider to be of great interest.

No

Comments

Corrections and Page Number

1

It would be necessary to update both the introduction and the theoretical framework of the references used. Physical activity and the sports area are scientific disciplines that have a lot of evidence and great academic production in recent years that will undoubtedly provide more up-to-date references to this article. This issue is repeated throughout the article, so a complete review and update of references is requested.

Thank you for the constructive comments. We have added relevant, latest references with publication years of 2021 and 2022. As for statistical facts from the Institute of Public Health, Malaysia, they are from the latest 2019 National Health and Morbidity Survey (NHMS). The next NHMS will be conducted in year 2024. Following references have been added into the introduction, theory building and discussion sections.

·         Anić et al., 2021

·         Kim, 2022

·         Kallio et al., 2021

·         Poveda-López et al., 2022

·         Redig et al., 2022

·         Stevens et al., 2021

2

With respect to the method, it would be interesting to know how the sample was selected, perhaps there are tables that are of little interest given the comments that derive from them, it being possible to consider removing them, the same thing happens with the results section, at times the information described is repeated with which appears in the tables.

Thank you; we agree that additional information about the sampling technique is necessary for future studies to test the framework with different populations. Thus, we have added more detail explanations (kindly refer to page 10, line 399-405 and page 10, line 413-414). Redundant information (page 10, line 406-407) and repetitive information (page 10, line 409-410) have been removed.

3

In the discussion, the authors are again requested to update the measurement of possible references and adapt the entire article with the tables and references to the format of the journal.

For updated (latest) references, kindly refer to point 1. We have reviewed literatures on social identification process in physical activity. Latest papers have discussed/investigated on group (in-group members) identification (instead of identification to a fitness leader/trainer), and effects of social identification/identity on continuity of PA participation, exercise adherence as well as turnover intention. We hope with our findings, more studies will be carried out to investigate  self-identification to a fitness trainer and related issues in sports psychology.

4

I consider it of great interest that the authors add a section on the implications of the study and limitations, since I consider that it is necessary for the reader to know the application of the work as well as how to face the limitations it presents.

Section 6 on study implication has been added accordingly (kindly refer to page 17 and 18, line 632-654), followed by Section 7 which is on Limitation of Study.

Reviewer 2 Report

Authors performed an interesting study using a mix methodology to explore a framework based on three major psychological theories: the Self-Determination Theory, the Social Comparison Theory and the Social Identity Theory. Each theory were well explained with enough details to make them understandable by audience less familiar with the field, which appears to me essential. Addition of Figure 1 helps a lot to understand the topic and make easy the reading. Authors explained very well where their hypothesis are coming from and established a good parallel with physical appearance through the study of behavioural triggers induced by exposure to physical activity. Statistics are also simple and easy to understand with separate tables for each analysis, compared to several articles who included monster tables with all variables (making difficult to catch the home message). The main issue emerging for me in this study was the fact that authors estimated the "intentions" not a fact/true action. The good fact is, authors presented themselves this situation as their major limitation. According to their discussion: the primary dependent variable of this study’s model is categorised under a behavioural intention, not an actual action. A behavioural intention does not necessarily turn into actual action in real-life practice. If they are aware of this problem which may affects potential generalization of their findings, i have nothing more to request for this manuscript. It will be a good reading!

Author Response

Thank you esteemed reviewer. 

Cover page with Errata Sheet: Response to Reviewer 2’ Comments and Correction

Article Title: “Gym Goer’s Identification to Physically Attractive Fitness Trainer and their Intention to Exercise”

Reviewer #2

Authors performed an interesting study using a mix methodology to explore a framework based on three major psychological theories: The Self-Determination Theory, the Social Comparison Theory and the Social Identity Theory. Each theory was well explained with enough details to make them understandable by audience less familiar with the field, which appears to me essential. Addition of Figure 1 helps a lot to understand the topic and make easy the reading. Authors explained very well where their hypothesis are coming from and established a good parallel with physical appearance through the study of behavioural triggers induced by exposure to physical activity. Statistics are also simple and easy to understand with separate tables for each analysis, compared to several articles who included monster tables with all variables (making difficult to catch the home message).

No

Comments

Corrections and Page Number

1

The main issue emerging for me in this study was the fact that authors estimated the "intentions" not a fact/true action. The good fact is, authors presented themselves this situation as their major limitation. According to their discussion: the primary dependent variable of this study’s model is categorised under a behavioural intention, not an actual action. A behavioural intention does not necessarily turn into actual action in real-life practice. If they are aware of this problem which may affects potential generalization of their findings, i have nothing more to request for this manuscript. It will be a good reading!

Thank you for the meaningful comments. We have included additional statements, including the precaution in generalisation of this study’s findings (kindly refer to page 18, line 651-661 and line 666-668).

Reviewer 3 Report

This manuscript was designed to understand this phenomenon better in the fitness industry. Relying on the Social Comparison Theory and the Social Identity Theory, identification with a physically attractive fitness trainer was posited to have a strong mediating effect on the relationship between convincing appearance, weight management motive, and gym goers' intention to sign-up for an exercise program. Overall, I believe the manuscript’s structure needs to be revised.    

Abstract

I would suggest the authors add the primary outcomes in the abstract

Introduction

Please provide the hypothesis of the study

I am not sure topic 2 “Theory building” is necessary. I would suggest shortening this part of the manuscript and maybe gathering with the Introduction Section.   

 Discussion Section

The authors reported in the manuscript approximately 12 tables showing their findings. However, the authors discuss such a finding in only two paragraphs. I believe it is very unproportionable and should be revised. 

Author Response

Thank you esteemed reviewer. 

Cover page with Errata Sheet: Responses to Reviewer 3’s Comments and Corrections

Article Title: “Gym Goer’s Identification to Physically Attractive Fitness Trainer and their Intention to Exercise”

Reviewer #3

This manuscript was designed to understand this phenomenon better in the fitness industry. Relying on the Social Comparison Theory and the Social Identity Theory, identification with a physically attractive fitness trainer was posited to have a strong mediating effect on the relationship between convincing appearance, weight management motive, and gym goers' intention to sign-up for an exercise program. Overall, I believe the manuscript’s structure needs to be revised.   

No

Comments

Corrections and Page Number

1

Abstract: I would suggest the authors add the primary outcomes in the abstract

Thank you for your constructive comments. Primary outcomes have been included in the abstract. Few sentences have been amended to better describe the study-findings (kindly refer to page 1).

2

Introduction: Please provide the hypothesis of the study

Brief description on the hypotheses has been included in Introduction section (kindly refer to page 3, line 89 to 95), while full list of all hypotheses is provided in section 2: Theory Building, precisely page 7 and 8, line 328 to 356.

3

I am not sure topic 2 “Theory building” is necessary. I would suggest shortening this part of the manuscript and maybe gathering with the Introduction Section.  

Thank you for the suggestion. We understand your concern as too lengthy section may be confusing to read. Section 2: Theory Building is 5 pages and half-length which summarizes all relevant theories that build the hypotheses. We believe inclusion of this section is necessary, particularly for readers who have no to limited knowledge about the three fundamental theories (SDT, Social Identity Theory and Social Comparison Theory) used in this study. As for the introduction, it encompasses the study background, research objectives and gaps that this study aims to fill-in. We have cut down some parts to shorten this section according to your suggestion. (page 4: line 179-183; page 5: line 213-214, line 217-218; page 6, line 237-239).

4

Discussion Section: The authors reported in the manuscript approximately 12 tables showing their findings. However, the authors discuss such a finding in only two paragraphs. I believe it is very unproportionable and should be revised.

We agree on this suggestion. We have added more explanation on the Discussion Section and expanded the discourse based on previous findings with reference to the findings (kindly refer to page 16, line 538-550, line 539-543, line 561-565 and line 573-582; page 17, line 569-601, line 604-614, ).